# Spectral Control by Silver Nanoparticle-Based Metasurfaces for Mitigation of UV Degradation in Perovskite Solar Cells

**DOI:** 10.3390/nano14191582

**Published:** 2024-09-30

**Authors:** Silvia Delgado-Rodríguez, Eva Jaldo Serrano, Mahmoud H. Elshorbagy, Javier Alda, Gonzalo del Pozo, Alexander Cuadrado

**Affiliations:** 1Departamento de Tecnología Electrónica, Universidad Rey Juan Carlos, 28933 Madrid, Spain; eva.jaldo@urjc.es (E.J.S.); gonzalo.delpozo@urjc.es (G.d.P.); 2Physics Department, Faculty of Science, Minia University, El Minia 61519, Egypt; mahmouha@ucm.es; 3Departamento de Óptica, Universidad Complutense de Madrid, 28037 Madrid, Spain; javier.alda@ucm.es

**Keywords:** perovskite solar cells, UV degradation, metasurfaces, core–shell nanoparticles, computational electromagnetism

## Abstract

Perovskite solar cells are considered to be one of the most promising solar cell designs in terms of photovoltaic efficiency. However, their practical deployment is strongly affected by their short lifetimes, mostly caused by environmental conditions and UV degradation. In this contribution, we present a metasurface made of silver nanoparticles used as a UV filter on a perovskite solar cell. The UV-blocking layer was fabricated and morphologically and compositionally analyzed. Its optical response, in terms of optical transmission, was also experimentally measured. These results were compared with simulations made through the use of a well-proven computational electromagnetism model. After analyzing the discrepancies between the experimental and simulated results and checking those obtained from electron beam microscopy and electron dispersion spectroscopy, we could see that a residue from fabrication, sodium citrate, strongly modified the optical response of the system, generating a redshift of about 50 nm. Then, we proposed and simulated the optical behavior of core–shell nanoparticles made of silver and silica. The calculated spectral absorption at the active perovskite layer shows how the appropriate selection of the geometrical parameters of these core–shell particles is able to tune the absorption at the active layer by removing a significant portion of the UV band and reducing the absorption of the active layer from 90% to 5% at a resonance wavelength of 403 nm.

## 1. Introduction

In recent years, third-generation perovskite-based solar cells (PSCs) have emerged as a promising technology in the field of photovoltaics due to their high efficiency, low manufacturing cost, and easy solution-based processing methods [1]. The certified maximum efficiency of PSCs has reached 26.1% [2], and they have been presented as an alternative to silicon-based technologies. 

Despite these advantages, PSCs still face challenges in terms of being positioned as competitors in the energy market, primarily due to their low operational lifespan [3]. The materials used in PSCs are sensitive to external agents, primarily moisture [4,5], oxygen [6,7], temperature [8,9,10], and ultraviolet (UV) light [11,12]. To mitigate the effects of the first two, techniques such as encapsulation can be employed to isolate cells from these agents [13,14]. However, a cell operating under normal conditions in direct sunlight will inevitably be exposed to UV light.

Despite the detrimental impact of UV light on PSCs, the underlying physics and chemistry of this degradation have not yet been fully elucidated. Several mechanisms have been proposed in the literature to elucidate the processes responsible for this degradation. The decomposition of the perovskite layer into its precursors, such as PbI_2_ or CH_3_NH_3_I, explains this type of degradation [15,16]. However, it appears that the mechanism that most affects the overall degradation of PSCs is due to the photocatalytic effect on the metallic oxide, which is typically used as an electron transport layer (ETL) in PSCs with regular architectures [12]. Specifically, the bandgap energy of TiO_2_ at the anatase phase, the most used material in terms of ETL films, has a value of 3.2 eV (410 nm), with absorption falling mostly in the UV spectrum. Upon exposure to UV radiation, TiO_2_ becomes excited, leading to the creation of electron–hole pairs, using photons with energy larger than it bandgap. Subsequently, holes generated in TiO_2_ react with surface-absorbed oxygen, resulting in the formation of deep traps, which promote charge recombination [17,18]. By efficiently filtering UV radiation, detrimental effects will be mitigated, allowing for the fabrication of perovskite solar cells with longer lifetimes.

Considering the solar spectrum, the irradiance of the UV region increases as it approaches the visible spectrum. Thus, filtering the UV region near 410 nm will largely eliminate the detrimental effects produced on the ETL of TiO_2_. Some recent advances on this topic adapt passivation layers on the ETL interface or mixed [19,20,21]. However, the standard protocol for achieving high-efficiency perovskite cells may be affected by adding more complexity to the layer structure. Furthermore, some designs could lead to transport losses and impact the final performance of the cell. As far as UV degradation is an optical problem, it can be resolved through optical design. For example, nanostructures that can modify the spectrum that reaches the active layer can be placed on already-fabricated devices [22,23]. However, these designs also have a negative effect on the visible part and may introduce reflectance, absorption losses, or both. One more criterion that should be considered in the design is to maintain the mechanical flexibility of the solar cell.

In recent years, nanophotonics has undergone a significant evolution, enabling the generation of metasurfaces and metamaterials whose optical responses can be precisely designed. A fundamental element in the generation of these structures is nanoparticles (NPs), which exhibit unique optical properties related to their size, shape, and material properties. This article focuses on the use of metal nanoparticles that show a characteristic localized plasmon resonance (LPR). This type of resonant response presents a high optical absorption with a narrow bandwidth, which can be spectrally tuned, allowing for the development of thin optical filters for specific spectral ranges.

The type of metal used and particle size determine the absorption band of these resonances, almost covering the whole electromagnetic spectrum [24,25]. In this sense, silver nanoparticles, which have a strong and sharp plasmonic resonance in the UV region, are an excellent choice for several applications, including this case study. A thin layer containing periodically or randomly distributed silver nanoparticles, which behaves as a metasurface with custom optical response, is one type of these filters. This metasurface can be attached to optical adhesives and placed directly over the transparent side of already-fabricated solar cell devices without affecting their fabrication protocol. The key to this design is to find a material and geometry for this metasurface that works as a perfect UV absorber with maximum alignment to TiO_2_ and also enables it to be as transparent as possible in the visible region. In the design and fabrication, we will follow strategies that adapt low-cost and easy processing methods to keep the final fabrication process efficient.

In this contribution, we focus on the ability of silver nanoparticle-based metasurfaces to generate UV filters. The reason for using silver is based on the strong plasmonic resonance in the UV portion of the spectrum. Ideally, we should remove the short wavelength responsible for the degradation of solar cells, maintaining an unaltered visible spectrum. However, when considering the efficiency of the charge carrier generation mechanism in terms of the short-circuit current—which is directly dependent on wavelength—we can see that the presence of some attenuation in the blue portion of the spectrum is not significant. Therefore, the extension of lifetime through UV filtering helps to outbalance the possible reduction in total efficiency.

After this introductory section, Section 2 outlines the optical behavior of the nanoparticles using a simulation model performed using the Finite Element Model (FEM) software Comsol Multiphysics 6.2. The simulations evaluate the optical response of a single nanoparticle in terms of its size and surrounding material. Section 3 describes the fabrication via a drop-out method and its subsequent optical characterization. The structural and composition analysis of the fabricated metasurfaces allows us to adjust the model by including the effects of the solvent on the local refractive index variation. One of these effects is the redshift of the spectral absorption that occurs when the particle is surrounded by a dielectric layer. This behavior suggests the use of core–shell nanoparticles, where the absorption of silver is spectrally modulated by the dielectric shell. Section 4 shows the simulated responses of optimized filters considering periodic arrays of nanoparticles in perovskite-based solar cells, both as silver and as a silver-SiO_2_ core–shell nanoparticle. In addition, a filter design is proposed that shows a higher robustness in its optical response to the effects produced by possible solvent residues. Finally, Section 5 summarizes the findings of this contribution.

## 2. Modelling and Simulations

In the optical domain, the electromagnetic response of a nanoparticle strongly depends on the material of the nanoparticle (through its optical properties), its geometry, and the optical constants of the surrounding media. A simple way to characterize this optical performance is through the extinction scattering coefficient, $Q_{ext}$, that can be given in terms of the absorption coefficient, $Q_{abs}$, and the scattering coefficient, $Q_{sca}$ as [26,27,28]:
(1)$$Q_{ext}=Q_{abs}+Q_{sca},$$

These two terms are defined by the following relations [26,27,28]:
(2)Qabs=∰J→E→dvIA,
(3)Qsc=∯S→ndaIA,
where *dv* and *da* are the differential elements of volume and surface of the nanoparticle, respectively. $\vec{J}$ and $\vec{E}$ are the induced current density and the electric field vector along the nanoparticle, respectively. *n* is the normal vector pointing outwards and $\vec{S}$ is the Poynting vector. Finally, *I* and *A* are the incident irradiance and the transversal area of the particle, respectively.

For nanoparticles much smaller than wavelength, d<<l/10, the Rayleigh scattering regime is applicable and absorption is predominant, meaning that $Q_{ext}=Q_{abs}$. Additionally, the optical response is quite simple, and the nanoparticle behaves as a point dipole, where the electric field is enhanced close to the nanoparticle. When the diameter increases, $Q_{sca}$ also grows and eventually it becomes larger than $Q_{abs}$. This general behaviour also depends on the surrounding medium and the presence of a substrate. To show this effect, we calculated the scattering coefficients for silver nanoparticles and the associated electric field map. We chose silver because of its good plasmonic response in UV, making it a good candidate to block this part of the spectrum [29]. The simulation considers silver nanoparticles and glass substrates with optical properties extracted from reliable sources [28,30,31]. The nanoparticle and the substrate are placed far enough from the emission port to avoid computational artifacts. The model consists of a light source spectrally resembling solar illumination conditions, physical domains (air–nanoparticle–substrate), and perfect matching layers (PMLs) to prevent reflection from returning to the physical domains. The lateral sides are adjusted with periodic boundary conditions to account for the infinite nature of the structure and reducing it to only a unit cell that is computationally resolvable. The solution is obtained through a two-step simulation. In the first step, we account for the solar irradiance that reaches the bottom of the substrate after interacting with the metasurface and substrate. Then, the second step uses this irradiance to account for absorption in the active layer of the cell. Due to the geometrical symmetry, we only need one calculation using a polarized plane wave (*E* = (0, 1, 0) V/m in our case) normally incident on the structure. This is sufficient to account for real nonpolarized light, as the results will be the same for the two orthogonal polarization states: transverse electric and transverse magnetic. The physical domains are meshed with tetrahedral elements, while PMLs are meshed using prismatic elements. A fine mesh, with the minimum element size related to the wavelength, is used to accurately resolve Maxwell’s equations to obtain the optical parameters.

The real fabricated sample consists of randomly distributed nanoparticle agglomerations, which can be approximated using effective index models [32]. For simplicity, we use a small unit cell with a single nanoparticle that is periodically repeated on an infinite scale. This simplification generally affects the depth and broadening of the resonance peaks but has no effect on the peak position [33].

Figure 1a–d are calculated for a silver nanoparticle with a diameter of 20 nm immersed in air. In this case, the Rayleigh regime shows that absorption is much larger than scattering. The electric field pattern in Figure 1b reproduces the expected dipolar emission of the nanoparticle. The difference between Figure 1a,b and Figure 1c,d is the presence of a BK7 glass substrate (n = 1.55), which strongly reduces the extinction coefficient by about 40%. The resonance in air, located at λ = 360 nm, redshifts slightly to λ = 365 nm, when the substrate is included. However, the spectral width remains the same, around ∆λ = 20 nm. The electric field distribution is also affected by the substrate, weakening the dipole pattern and enhancing the field at the nanoparticle–substrate interface. 

When the nanoparticle’s diameter increases up to 80 nm, the scattering coefficient also increases and redshifts relative to the absorption coefficient. This is because the Rayleigh approximation is no longer valid, and a multipole resonance, explained through Mie scattering, appears. As a result, the extinction coefficient redshifts and reaches its maximum at 395 nm (see Figure 1e), showing a wider spectral response. Since we aim for large spectral absorption in the UV and high transmittance in the visible range, we need dominant absorption by the nanoparticle (in the Rayleigh regime) that does not overlap with the visible spectrum [24,25]. This is why the observed shift towards the visible spectrum and the reduction in the absorption efficiency make large-diameter silver nanoparticles on glass a poor choice as a UV filter. Therefore, we will focus our attention on silver nanoparticles with diameters around 20 nm.

## 3. Sample Preparation and Analysis

After considering the results obtained from our simulations, we prepared several samples using commercially available silver nanospheres having a diameter of 20 ± 4 nm.

Proper preparation of the glass substrate is crucial to obtain an adequate dispersion and adhesion of NPs onto it. To achieve this, the surface of the glass must be functionalized as follows. Initially, the substrate undergoes washing with deionized water, followed by acetone, deionized water again, ethanol, and, finally, deionized water. After this wet washing recipe, the substrate is dried in oven at 400 °C for one hour to remove any residual organic compounds. The next step is surface hydroxylation by adding hydroxyl (–OH) groups to the substrate, which is carried out by immersing it in a piranha acid solution for 30 min. This process also increases the roughness of the substrate, improving the adhesion of NPs. Piranha solution is prepared by mixing 30 mL of sulfuric acid (H_2_SO_4_) with 10 mL of hydrogen peroxide (H_2_O_2_). Then, the substrates undergo a silanization process by immersing them in a solution of APTES (3-aminopropyltriethoxysilane) in ethanol at a concentration of 1% by volume for 24 h. To finish the functionalization process, the substrate is rinsed with ethanol and dried in an oven at 110 °C for 24 h in air.

The silver NPs were deposited on the prepared substrate using the drop-out technique. For this purpose, 350 mL of a commercial dispersion (Merck Life Science S.L.U., Darmstadt, Germany, Silver, dispersion 730793) was poured onto the functionalized glass substrates. This dispersion consists of silver nanoparticles, 20 nm in particle size, dispersed at 0.02 mg/mL in an aqueous buffer that contains sodium citrate as a stabilizer. After that, the solvent was removed by heating the samples to 100 °C for one hour.

Although the drop-out technique is simple and fast, it does not allow for precise control of the quantity of nanoparticles and their dispersion on the surface. SEM images of the samples were taken to analyze the resulting surface. The distribution of the silver NPs is shown in Figure 2. Most of the NPs are embedded within an external material. An Energy-Dispersive Spectroscopy (EDS) analysis conducted during the measurement revealed several peaks for sodium, oxygen, and carbon, suggesting that the material adhering to the NP is sodium citrate (Na_3_C_6_H_5_O_7_). This compound is commonly used as a dispersion agent in commercial NP products. From the image, we can estimate that the deposited nanoparticles are 20 nm in diameter, and the average distance between them is about 122 nm.

## 4. Analysis and Discussion of the Optical Response

Simulation was performed with a periodic array of silver nanoparticles surrounded by air on a glass substrate. However, the electron microscopy analysis reveals that real conditions differ from those used in the simulations. The spatial distribution of NPs is not homogeneous, and some clusters of NPs appear on the substrate. Moreover, the presence of sodium citrate, which surrounds the NPs and fixes them to the glass, changes the refractive index around the NPs. To consider this effect, we included, in the computational electromagnetism model, a spherical shell with refraction index different from that of glass. We set this refractive index by fitting the optical response of the structure with the measured transmission of the sample. 

In Figure 3, we plotted the experimental spectral transmittance for the samples obtained through the process presented in Section 3 (see plot in dashed black line). This curve shows a minimum value at λ = 420 nm. Then, a simulation is made by changing the refractive index of the spherical shell to obtain the minimum transmittance at the same wavelength as the experiment. The best fit is obtained with a refractive index of n_Shell_ = 1.58 and diameter of 60 nm for the spherical shell. In the same plot, we present the spectral transmission for a silver NP without the sodium citrate spherical shell (see plot in dashed green line) that resonates and absorbs at λ = 358 nm. In Figure 3b,c, we show the electric field maps for the two resonances: at 358 nm where the NP is surrounded by air, and at 420 nm for the case of the sodium citrate spherical shell.

The results from the simulation (see Figure 3) demonstrate how the presence of sodium citrate shifts the absorption peak towards the visible spectrum. On the other hand, when comparing the experimental and simulated results, it is evident that the bandwidth is wider for the experiment, ∆λ = 60 nm, than for the simulation, ∆λ = 15 nm. Moreover, the experimental minimum transmittance, T_min_ = 0.83, is not as pronounced as in the simulated case, T_min_ = 0.62. One reason for the broadening and the higher transmission at the resonance is the random spatial localization of the NPs and the presence of clusters, where they interact differently than in the regularly spaced case. Also, from the electron microscopy, we can see that the sodium citrate coating is not homogeneous in thickness, adding more variability to the nominal case simulated through computational electromagnetism. All these deviations from the ideal case are responsible for a broadening of the resonance and the shallowness of the dip in the spectral transmission. However, both the simulation and experiment present the concept well, and this geometrical issue can be solved by applying other methods such as nanosphere lithography [34].

When applying the NP metasurface as a UV-blocking filter, it is important to compute the absorption within the perovskite layer. The simulation model contemplates a four-cation perovskite solar cell with a standard structure. Specifically, this structure includes a glass substrate, a 300 nm thick FTO layer that works as a transparent contact, an ETL based on two compact and mesoporous TiO_2_ layers with a thickness of 60 and 200 nm, respectively, a 500 nm perovskite active layer, an HTL based on a 240 nm Spiro-OMeTAD layer, and, finally, a metallic Au contact. Proportionally to the quantum efficiency of the process, each absorbed photon is transformed into a pair of charge carriers that generates the photo-induced current. 

Figure 4 shows the computed absorption for several cases of interest. The grey dotted line in Figure 4b represents the case of the solar cell without any additional elements. The blue dotted line is the absorption for the perovskite layer of the same cell, where we added a regular arrangement of silver NPs with a diameter of 20 nm and a spatial period of 40 nm. As expected, absorption shows a minimum at λ = 360 nm, affecting a band between 304 and 370 nm in the UV region. Unfortunately, the calculated absorption decreases around 4% in the visible range where the photovoltaic conversion is more efficient. Finally, the red solid line corresponds to the case of the same spatial arrangement of the NPs, but in this case, each silver nanosphere is coated with a spherical shell of a dielectric with n = 1.58 and a thickness of 20 nm, mimicking the effect of the sodium citrate. In this last case, the minimum in absorption occurs at λ = 410 nm. The situation for the sodium citrate-coated NPs generates absorption in the visible range that is clearly lower than for the perovskite alone, significantly affecting the efficiency of the solar cell.

The results previously obtained for the silver NPs suggest new design strategies for better UV blocking that do not jeopardize the overall performance of the solar cell. Our solution was inspired by the observed effect of the spherical shell surrounding the NPs. In fact, we propose the use of core–shell NPs, where the silver nanospheres are covered by a layer of a known dielectric. If this dielectric material is thick enough, it will be solely responsible for the optical response of the NPs, without additional disturbances from residues of the functionalization process or those present in the NP preservation preparation.

Our design uses silver nanospheres coated with silica, which are commercially available and solve the oxidation issues of silver nanospheres. The simulation starts with a silver core of 20 nm in diameter, coated with a 20 nm thick layer of silica. The core–shell NPs are then arranged periodically on a square lattice on the BK glass substrate, with a period of 80 nm. Figure 5 depicts the results obtained for this design. The absorption at the perovskite active layer is plotted as a grey dotted line. The thickness of this active layer is 500 nm. The case of the periodic core–shell NPs is represented by the blue line, showing a sharp minimum of absorption centered at λ = 392 nm. In the visible spectrum, the absorption is slightly lower than that of the regular solar cell due to the increased reflectance caused by the NPs. Finally, we simulated the effect of an additional layer of sodium citrate that could remain after the deposition process (red dashed line). The new resonance redshifts to λ = 403 nm with a spectral width of 20 nm, decreasing the absorptance in the visible band. As observed with the sodium citrate coating previously presented in Figure 4, there is a redshift of 10 nm when this residue is considered in the computational model. In Figure 5, we can also see how this type of core–shell NP enhances the electric field at the dielectric spherical shell, significantly diminishing the effect of the material surrounding the core–shell NPs.

An important issue when adding new absorbing structures in solar cells is the potential reduction in the efficiency of the cell. This efficiency is strongly associated with the short-circuit current, *J_SC_*, which can be calculated using the equation [35,36]:
(4)$$J_{SC}=\int q\frac{\lambda }{hc}A\left(\lambda \right)\phi_{AM1.5}\left(\lambda \right)d\lambda ,$$
where *q* is the electron charge, *c* is the speed of light in a vacuum, *h* is the Planck’s constant, *ϕ_AM_*_1.5_ (*λ*) is the standard solar spectral irradiance, and $A\left(\lambda \right)$ is the absorption at the active layer of the cell. The previous equation assumes a quantum efficiency of 1. The spectral absorption at the perovskite layer is shown in Figure 4b and Figure 5b, allowing for a simple calculation of the short-circuit current. Regarding the effect of the filters proposed in this contribution, we calculated the ratio between the *J_SC_* obtained when adding the filter and the *J_SC_* generated by the perovskite cell without a filter, ρ*_filter_* = *J_SC_*_,*filter*_/*J_SC_*_,*perovs*_.

The results of this calculation confirm that the effect of the filter is almost negligible: ρ_NPAg_ = 98.60% and ρ_NPAgSiO2_ = 98.64%. However, the presence of the citrate layer affects the efficiency to a greater extent: ρ_NPAg,citrate_ = 88.46% and ρ_ΝPAgSiO2,citrate_ = 91.05%. These numbers suggest that, from the point of view of cell efficiency, the filter does not significantly alter it once the citrate layer is removed. In fact, core–shell NPs are usually immersed in volatile solvents different from sodium citrate. Therefore, mitigating the degradation of the cell by filtering some UV portion of the spectrum is achieved at an affordable cost in terms of efficiency reduction (less than 2%). This means that the total energy collected by the cell is greater when extending the lifetime of the element using the spectral control strategies presented here.

## 5. Conclusions

In this work, we presented a simple method to develop a UV filter based on silver nanoparticles. These NPs were deposited by dropping a commercial Ag dispersion onto a functionalized glass substrate. For a more precise deposition of the NPs and their corresponding absorption spectrum, we suggest the use of alternative techniques, such as pulsed laser deposition (PLD) or nanosphere lithography. SEM characterization shows a silver dispersion of 20 nm NPs with an average separation of 122 nm, surrounded by citrate residue, which changes the resonance response of the UV filter. Using COMSOL Multiphysics simulations, we estimated the refractive index of this residue to be n_Shell_ = 1.58. To achieve this, we used the measured spectral transmission data.

To avoid the effect of citrate residue on the plasmon resonance of the NPs, we propose a new filter based on a core–shell structure, made of Ag nanoparticles 20 nm in diameter, surrounded by an additional 20 nm of silica. The results of the simulations show a narrow absorption centered at λ = 392 nm. This new structure is less sensitive to the presence of citrate residues, with a small redshift in the absorption peak to λ = 403 nm and a width of only 20 nm. This wavelength is close to the maximum absorption of TiO_2_, a material widely used in the fabrication of perovskite solar cells and one of the main causes of their degradation. When calculating the effect of the filter on the efficiency of the solar cell, we found that the NP layer, whether silver or core–shell particles, does not significantly change the short-circuit current, which remains at around 99% of the value obtained with the perovskite cell without the filter. Therefore, we propose this type of filter to extend the lifetime of the perovskite cell, increasing the total energy collected by the cell along its lifetime. The fabrication solutions are easy to implement and maintain the low-cost philosophy of this type of device. Thus, we propose this type of filter to increase the robustness of perovskite cells, using techniques that are easy to implement while keeping the low-cost philosophy of this type of device.

## Figures and Tables

**Figure 1 nanomaterials-14-01582-f001:**
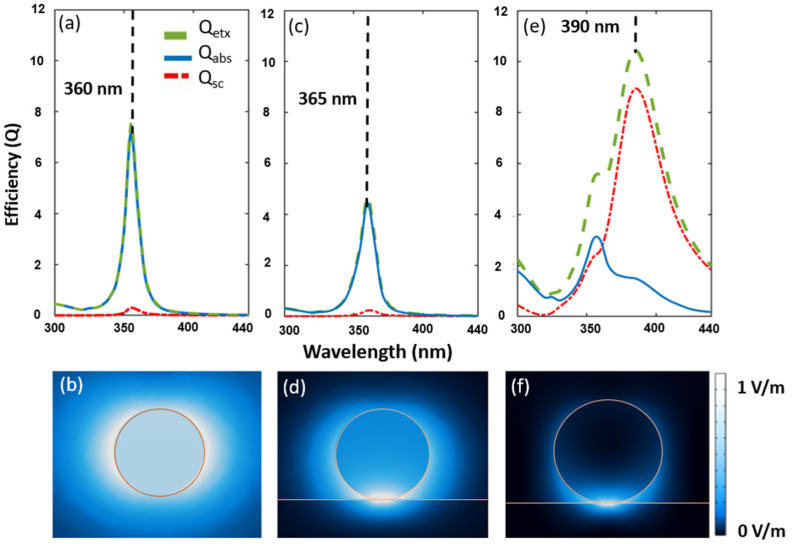
Calculation of the absorption coefficients *Q_ext_*, *Q_abs_*, and *Q_sc_*, and the electric field distributions for several cases involving Ag NP. We show the simulation results for (**a**) a 20 nm Ag NPs surrounded by air, (**c**) a 20 nm Ag NPs surrounded by air on a glass substrate (**e**) an 80 nm Ag NPs surrounded by air on glass substrate. (**b**,**d**,**f**) are the normalized electric field distributions for the cases in (**a**,**c**,**e**), respectively.

**Figure 2 nanomaterials-14-01582-f002:**
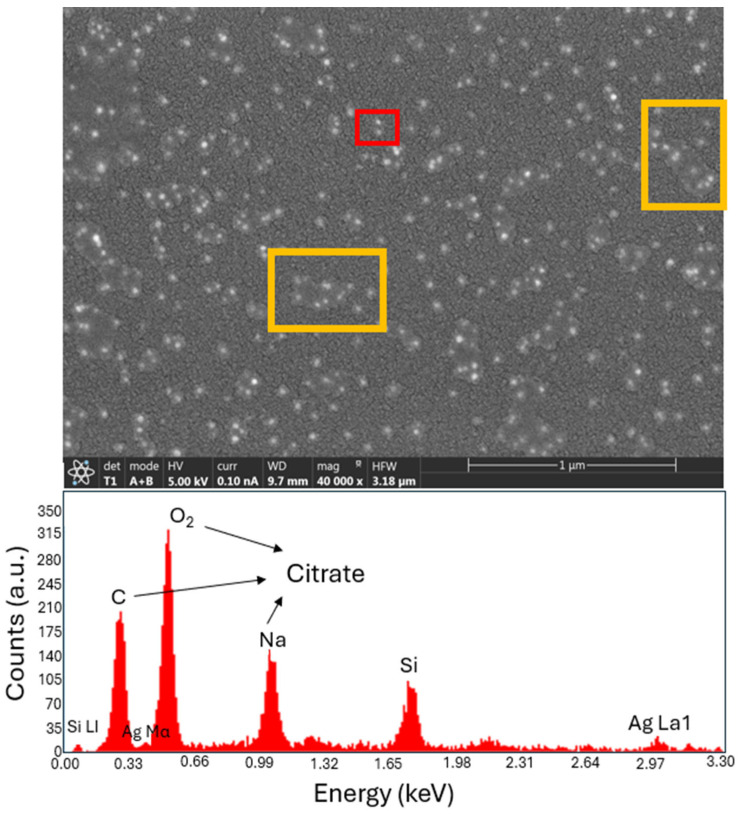
(**Top**): Electron microscope photography of the sample showing the presence of silver nanoparticles on the glass substrate, and citrate residue (boxes). The NPs are embedded within a medium. (**Bottom**): EDS results for the regions marked in the top photography. This spectrogram shows the presence of Na, O_2_, and C, that are part of sodium citrate.

**Figure 3 nanomaterials-14-01582-f003:**
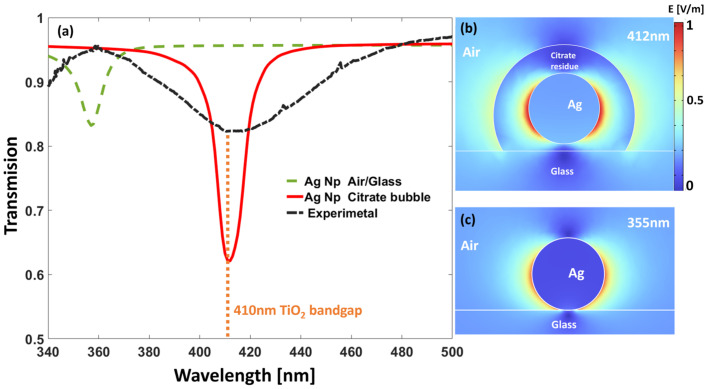
(**a**) Spectral transmittance obtained experimentally from the samples (in dashed black line), and simulated for a periodic arrangement with period 122 nm for a collection of 20 nm silver spherical NPs located on a glass substrate (in dashed green) and for the same collection of NPs surrounded by a spherical cell of thickness 20 nm of a dielectric material having an index of refraction n = 1.58 (equivalent to the sodium citrate spherical shell). (**b**,**c**) Maps of the electric field modulus for the case of the equivalent sodium citrate spherical shell, and the silver NP surrounded by air, respectively. Both cases have the NPs located on a BK7 glass substrate.

**Figure 4 nanomaterials-14-01582-f004:**
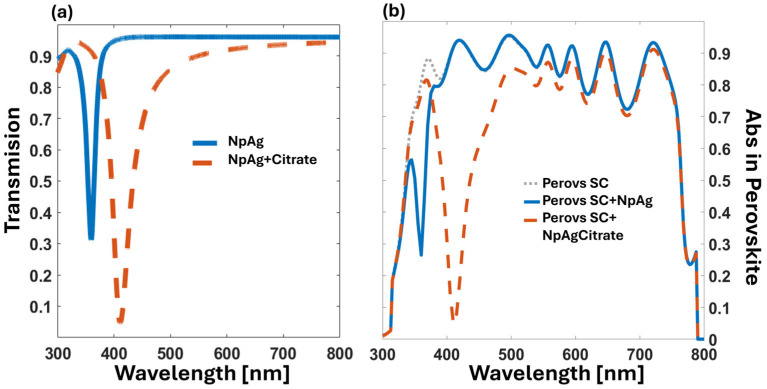
(**a**) Simulated optical transmission of the UV filter based on Ag nanoparticle. (**b**) Simulated spectral absorptance at the filtered perovskite active layer for the cases of interest. The grey dotted line is for the solar cell alone. To both figures, the blue line is for a regular arrangement of silver NPs, 20 nm in diameter and spatial period of 40 nm, while the red dashed line is for the same NPs but coated with a dielectric layer which simulates the sodium citrate around the silver NPs.

**Figure 5 nanomaterials-14-01582-f005:**
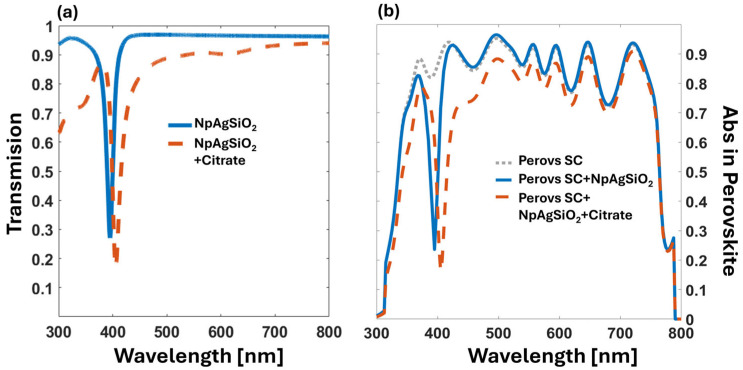
(**a**) Simulated optical transmission of the UV filter based on silver–silica core–shell nanoparticle. (**b**) Simulated spectral absorption at the active perovskite layer showing the result for the regular cell (grey dotted line). To both figures, the blue line is for a periodic arrangement of core–shell silver-silica NPs, and the red dashed line is related to a periodic arrangement is coated with a material that reproduces the results of sodium citrate residue.

## Data Availability

Data will be made available on request.

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
