# Peer review of "Spectral Control by Silver Nanoparticle-Based Metasurfaces for Mitigation of UV Degradation in Perovskite Solar Cells"

_nanomaterials, 2024, doi:10.3390/nano14191582_

Round 1

Reviewer 1 Report

Comments and Suggestions for Authors

This work present a metasurface made of silver nanoparticles as UV filter on perovskite solar cell. This work is interesting and the presented from is fine. Therefore, I recommend to accept this manuscript after addressing the following issues.

1. The keywords include perovskite solar cells. The author should assemble the perovskite solar cells with UV filter based on silver nanoparticles and measure their photovoltaic properties and working stability.  

2. As can be seen from the light absorption curve, the addition of silver nanoparticles will lead to a decrease in light absorption ability, which will reduce the photoelectric conversion efficiency of the solar cell? The author should consider this impact.

Reviewer 2 Report

Comments and Suggestions for Authors

Comments:

The authors presented a simple method to develop an UV filter based on silver nanoparticles for mitigation of UV degradation in Perovskite Solar Cells. These NPs were deposited by dropping a commercial Ag dispersion onto a functionalized glass substrate. Therefore, they propose this type of filter to increase the life-time of perovskite cells, using techniques that are easy to implement and keeping the low-cost philosophy of this type of device. The paper could be further considered for publication with addressing the following issues:

1.     The authors are developing something Ag nanoparticles for filter in perovskite solar cells, but the ITO or even the TiO2 already absorbs part of the uv light. Anything that absorbs above the absorption of these will cause a decrease in the solar cell efficiency. So, I would like, first, a better motivation to use Ag nanoparticles as uv filters in perovskite solar cells.

2.     In figure 1, authors should state which simulated results are presented, especially in a,c, and e. Also, put the name of y axis.

3.     Author should also comment about the limitations and approximations taken into account in COMSOL.

4.     I would like to see a perovskite solar cell made with this filter.

5.     How the silver nanoparticles affect wettability of the substrate and its influence on the perovskite film formation?

Reviewer 3 Report

Comments and Suggestions for Authors

In the attachment.
